# Learning Knowledge-Enhanced Contextual Language Representations for Domain Natural Language Understanding

**Taolin Zhang**[1,2][*] **Ruyao Xu**[1][*] **Chengyu Wang**[2], **Zhongjie Duan**[1], **Cen Chen**[1][†],
**Minghui Qiu**[2], **Dawei Cheng**[3], **Xiaofeng He**[1][†] **Weining Qian**[1]

[1] East China Normal University, Shanghai, China
[2] Alibaba Group, Hangzhou, China [3] Tongji University, Shanghai, China
{zhangtaolin.ztl,chengyu.wcy,minghui.qmh}@alibaba-inc.com
{ryxu,zjduan}@stu.ecnu.edu.cn, {cenchen,wnqian}@dase.ecnu.edu.cn
hexf@cs.ecnu.edu.cn, dcheng@tongji.edu.cn

## Abstract

Knowledge-Enhanced Pre-trained Language Models (KEPLMs) improve the performance of various downstream NLP tasks by injecting knowledge facts from large-scale Knowledge Graphs (KGs). However, existing methods for pre-training KEPLMs with relational triples are difficult to be adapted to close domains due to the lack of sufficient domain graph semantics. In this paper, we propose a Knowledge-enhanced lANGuAge Representation learning framework for various clOsed dOmains (KANGAROO) via capturing the implicit graph structure among the entities. Specifically, since the entity coverage rates of closed-domain KGs can be relatively low and may exhibit the *global sparsity* phenomenon for knowledge injection, we consider not only the shallow relational representations of triples but also the hyperbolic embeddings of deep hierarchical entity-class structures for effective knowledge fusion. Moreover, as two closed-domain entities under the same entity-class often have *locally dense* neighbor subgraphs counted by max point biconnected component, we further propose a data augmentation strategy based on contrastive learning over subgraphs to construct hard negative samples of higher quality. It makes the underlying KELPMs better distinguish the semantics of these neighboring entities to further complement the global semantic sparsity. In the experiments, we evaluate KANGAROO over various knowledge-aware and general NLP tasks in both full and few-shot learning settings, outperforming various KEPLM training paradigms performance in closed-domains significantly. [1]

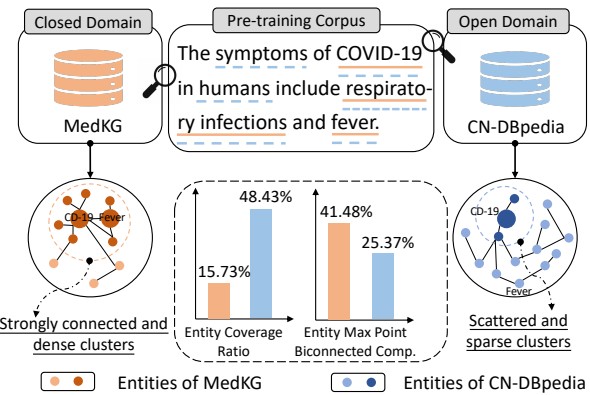

Figure 1: Comparison of statistics between closed-domain and open-domain KGs (taking CN-DBPedia (Xu et al., 2017) and a medical KG as an example). Closed-domain KGs have lower entity coverage ratios over text corpora (*global sparsity*). Entities are more densely inter-connected in closed-domain KGs (*local density*) [2]. (Best viewed in color.)

## 1 Introduction

The performance of downstream tasks (Wang et al., 2020) can be further improved by KEPLMs (Zhang et al., 2019; Peters et al., 2019; Liu et al., 2020a; Zhang et al., 2021a, 2022a) which leverage rich knowledge triples from KGs to enhance language representations. In the literature, most knowledge injection approaches for KEPLMs can be roughly categorized into two types: *knowledge embedding* and *joint learning*. (1) *Knowledge embedding*-based approaches aggregate representations of knowledge triples learned by KG embedding models with PLMs' contextual representations (Zhang et al., 2019; Peters et al., 2019; Su et al., 2021; Wu et al., 2023). (2) *Joint learning*-based methods convert knowledge triples into pre-training sentences without introducing other parameters for knowledge encoders (Sun et al., 2020; Wang et al., 2021). These works mainly focus on building KEPLMs for the open domain based on large-scale KGs (Vulic et al., 2020; Lai et al., 2021; Zhang et al., 2022c).

---

[*]T. Zhang and R. Xu contributed equally to this work.
[†]Co-corresponding authors.

[1]All the codes and model checkpoints have been released to public in the EasyNLP framework (Wang et al., 2022). URL: https://github.com/alibaba/EasyNLP.

[2]The detailed analysis of entity coverage ratios and max point biconnected component is described in Sec. 2

Despite the success, these approaches for building open-domain KEPLMs can hardly be migrated directly to closed domains because they lack the in-depth modeling of the characteristics of closed-domain KGs (Cheng et al., 2015; Kazemi and Poole, 2018; Vashishth et al., 2020). As in Figure 1, the coverage ratio of KG entities w.r.t. plain texts is significantly lower in closed domains than in open domains, showing that there exists a *global sparsity* phenomenon for domain knowledge injection. This means injecting the retrieved few relevant triples directly to PLMs may not be sufficient for closed domains. We further notice that, in closed-domain KGs, the ratios of maximum-point biconnected components are much higher, which means that entities under the same entity-class in these KGs are more densely interconnected and exhibit a *local density* property. Hence, the semantics of these entities are highly similar, making the underlying KEPLMs difficult to capture the differences. Yet a few approaches employ continual pre-training over domain-specific corpora (Beltagy et al., 2019; Peng et al., 2019; Lee et al., 2020), or devise pre-training objectives over in-domain KGs to capture the unique domain semantics (which requires rich domain expertise) (Liu et al., 2020b; He et al., 2020). Therefore, there is a lack of a simple but effective unified framework for learning KEPLMs for various closed domains.

To overcome the above-mentioned issues, we devise the following two components in a unified framework named KANGAROO. It aggregates the above implicit structural characteristics of closed-domain KGs into KEPLM pre-training:

- **Hyperbolic Knowledge-aware Aggregator:** Due to the semantic deficiency caused by the *global sparsity* phenomenon, we utilize the Poincaré ball model (Nickel and Kiela, 2017) to obtain the hyperbolic embeddings of entities based on the entity-class hierarchies in closed-domain KGs to supplement the semantic information of target entities recognized from the pre-training corpus. It not only captures richer semantic connections among triples but also implicit graph structural information of closed-domain KGs to alleviate the sparsity of global semantics.

- **Multi-Level Knowledge-aware Augmenter:** As for the *local density* property of closed-domain KGs, we employ the contrastive learning framework (Hadsell et al., 2006; van den Oord et al., 2018) to better capture fine-grained semantic differences of neighbor entities under the same entity-class structure and thus further alleviate global sparsity. Specifically, we focus on constructing high-quality multi-level negative samples of knowledge triples based on the relation paths in closed-domain KGs around target entities. By using the proposed approach, the difficulty of being classified of various negative samples is largely increased by searching within the max point biconnected components of the KG subgraphs. This method enhances the robustness of domain representations and makes the model distinguish the subtle semantic differences better.

In the experiments, we compare KANGAROO against various mainstream knowledge injection paradigms for pre-training KEPLMs over two closed domains (i.e., medical and finance). The results show that we gain consistent improvement in both full and few-shot learning settings for various knowledge-intensive and general NLP tasks.

## 2 Analysis of Closed-Domain KGs

In this section, we analyze the data distributions of open and closed-domain KGs in detail. Specifically, we employ OpenKG [2] as the data source to construct a medical KG, denoted as MedKG. In addition, a financial KG (denoted as FinKG) is constructed from the structured data source from an authoritative financial company in China[3]. As for the open domain, CN-DBpedia[4] is employed for further data analysis, which is the largest open source Chinese KG constructed from various Chinese encyclopedia sources such as Wikipedia.

To illustrate the difference between open and closed-domain KGs, we give five basic indicators (Cheng et al., 2015), which are described in detail in Appendix A due to the space limitation. From the statistics in Table 1, we can roughly draw the following two conclusions:

**Global Sparsity**. The small magnitude and the low coverage ratio lead to the global sparsity problem for closed-domain KGs. Here, data magnitude refers to the sizes of *Nodes* and *Edges*. The low

---

[2]http://openkg.cn/. The medical data is taken from a subset of OpenKG. See http://openkg.cn/dataset/disease-information.
[3]http://www.seek-data.com/
[4]http://www.openkg.cn/dataset/cndbpedia

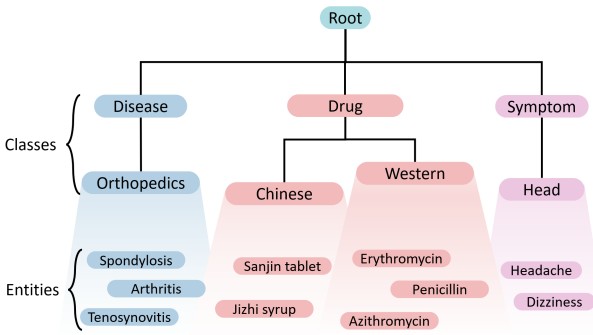

Figure 2: Illustration of the hierarchical structure of entities and classes in closed-domain KGs (e.g. MedKG).

| Statistics | Closed Domain | | Open Domain |
| | FinKG | MedKG | CN-DBpedia |
|---|---|---|---|
| #Nodes | 9.4 e+3 | 4.4 e+4 | 3.0 e+7 |
| #Edges | 1.8 e+4 | 3.0 e+5 | 6.5 e+7 |
| %Coverage Ratio | 5.82% | 15.75% | 41.48% |
| %Max PBC | 46.86% | 48.43% | 25.37% |
| Subgraph Density | 2.1 e-4 | 1.5 e-4 | 1.1 e-7 |

Table 1: The statistics of open and closed-domain KGs.

entity coverage ratio causes the lack of enough external knowledge to be injected into the KEPLMs. Meanwhile, the perplexing domain terms are difficult to be covered by the original vocabulary of open-domain PLMs, which hurts the ability of semantic understanding for closed-domain KEPLMs due to the out-of-vocabulary problem. From Figure 2, we can see that the closed-domain KGs naturally contain the hierarchical structure of entities and classes. **To tackle the insufficient semantic problem due to entity coverage, we inject the domain KGs' tree structures rather than the embeddings of entities alone into the KEPLMs. Local Density**. It refers to the locally strong connectivity and high density of closed-domain KGs, which are concluded by the statistics of maximum point biconnected components and subgraph density. Compared to the number of the surrounding entities and the multi-hop structures in the sub-graph of target entities in open-domain KGs, we find that target entities in closed-domain KGs are particularly closely and densely connected to the surrounding related neighboring entities based on the statistics of *Max PBC* and *Subgraph Density*. Hence, these entities share similar semantics, which the differences are difficult for the model to learn. **We construct more robust, hard negative samples for deep contrastive learning to learn the fine-grained semantic differences of target entities in closed-domain KGs to further alleviate the global sparsity problem.**

## 3  KANGAROO Framework

In this section, we introduce the various modules of the model in detail and the notations are described in Appendix B.5 due to the space limitation. The whole model architecture is shown in Figure 3.

### 3.1  Hyperbolic Knowledge-aware Aggregator

In this section, we describe how to learn the hyperbolic entity embedding and aggregate the positive triples' representations to alleviate the global sparsity phenomenon in closed-domain KGs.

#### 3.1.1  Learning Hyperbolic Entity Embedding

As discovered previously, the embedding algorithms in the Euclidean space such as (Bordes et al., 2013) are difficult to model complex patterns due to the dimension of the embedding space. Inspired by the Poincaré ball model (Nickel and Kiela, 2017), the hyperbolic space has a stronger representational capacity for hierarchical structure due to the reconstruction effectiveness. To make up for the global semantic deficiency of closed domains, we employ the Poincaré ball model to learn structural and semantic representations simultaneously based on the hierarchical entity-class structure. The distance between two entities $(e_i, e_j)$ is:

$$d(e_i, e_j) = \mathcal{F}_h\left(1 + \frac{2\|\mathcal{H}(e_i) - \mathcal{H}(e_j)\|^2}{(1 - \|\mathcal{H}(e_i)\|^2)(1 - \|\mathcal{H}(e_j)\|^2)}\right) \quad (1)$$

where $\mathcal{H}(.)$ denotes the learned representation space of hyperbolic embeddings and $\mathcal{F}_h$ means the $arcosh$ function. We define $D = \{r(e_i, e_j)\}$ be the set of observed hyponymy relations between entities. Then we minimize the distance between related objects to obtain the hyperbolic embeddings:

$$\mathcal{L}(\theta) = \sum_{r(e_i, e_j) \in D} \log \frac{\exp(-d(e_i, e_j))}{\sum_{e'_j \in \phi} \exp\left(-d(e_i, e'_j)\right)} \quad (2)$$

where $\phi$ means $Neg(e_i) = \{e'_j | r(e_i, e'_j) \notin D\} \cup \{e_j\}$ and $\{e_j\}$ is the set of negative sampling for $e_i$. The entity class embedding of token $t_{je}^C$ can be formulated as $h_{c_j} = \mathcal{H}(C | t_{je}^C \in C) \in \mathbb{R}^{d_2}$.

#### 3.1.2  Domain Knowledge Encoder

This module is designed for encoding input tokens and entities as well as fusing their heterogeneous

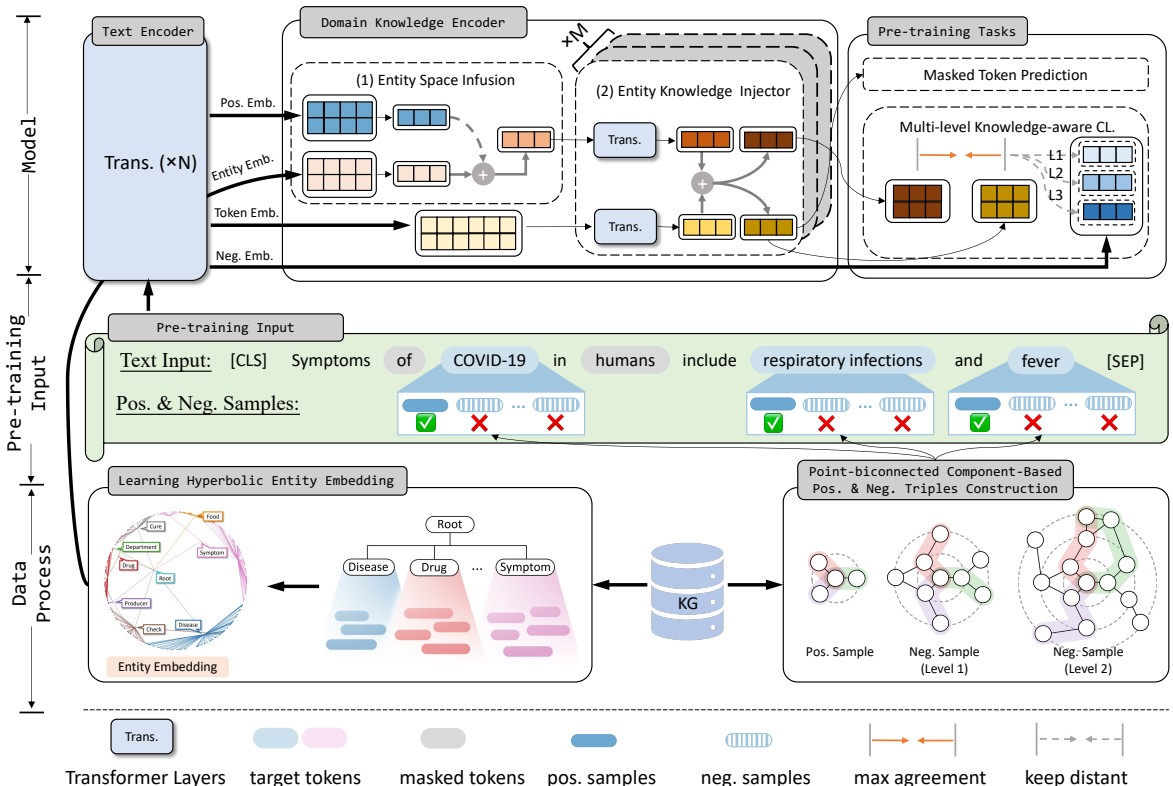

Figure 3: Model overview of KANGAROO. The *Hyperbolic Entity Class Embedding* module mainly leverages the hierarchical entity-class structure to provide more sufficient semantic knowledge. *Positive and Negative Triple Construction* can obtain negative samples of higher quality in multiple difficulty levels. (Best viewed in color.)

embeddings, containing two parts: *Entity Space Infusion* and *Entity Knowledge Injector*.

**Entity Space Infusion**. To integrate hyperbolic embeddings into contextual representations, we inject the entity class embedding $h_{c_j}$ into the entity representation $h_{p_j}$ by concatenation:

$$\hat{h}_{e_j} = \sigma([h_{c_j}\|h_{p_j}]W_f + b_f) \quad (3)$$

$$h_{e_j} = \mathcal{LN}(\hat{h}_{e_j}W_l + b_l) \quad (4)$$

where $\sigma$ is activation function GELU (Hendrycks and Gimpel, 2016) and $\|$ means concatenation. $h_{p_j}$ is entity representation (See Section 3.2.1). $\mathcal{LN}$ is the LayerNorm fuction (Ba et al., 2016). $W_f \in \mathbb{R}^{(d_1+d_2)\times d_3}$, $b_f \in \mathbb{R}^{d_3}$, $W_l \in \mathbb{R}^{d_3 \times d_3}$ and $b_l \in \mathbb{R}^{d_3}$ are parameters to be trained.

**Entity Knowledge Injector**. It aims to fuse the heterogeneous features of entity embedding $\{h_{e_j}\}_{j=1}^m$ and textual token embedding $\{h_{t_i}\}_{i=1}^n$. To match relevant entities from the domain KGs, we adopt the entities that the number of overlapped words is larger than a threshold. We leverage the $M$-layer aggregators as knowledge injector to be able to integrate different levels of learned fusion results. In each aggregator, both embeddings are fed into a multi-headed self-attention layer denoted as $\mathcal{F}_m$:

$$\{h'_{e_j}\}_{j=1}^m, \{h'_{t_i}\}_{i=1}^n = \sum_{v=1}^M \mathcal{F}_m^v(\{h_{e_j}\}_{j=1}^m, \{h_{t_i}\}_{i=1}^n) \quad (5)$$

where $v$ means the $v_{th}$ layer. We inject entity embedding into context-aware representation and recapture them from the mixed representation:

$$\hat{h}'_i = \sigma(\hat{W}_t h'_{e_j} + \hat{W}_e h'_{ti} + \hat{b}) \quad (6)$$

$$\hat{h}'_{t_i} = \sigma(W_e \hat{h}'_i + b_t) \quad \hat{h}'_{e_j} = \sigma(W_t \hat{h}'_i + b_e) \quad (7)$$

where $\hat{W}_t \in \mathbb{R}^{d_1 \times d_4}$, $\hat{W}_e \in \mathbb{R}^{d_3 \times d_4}$, $\hat{b} \in \mathbb{R}^{d_4}$, $W_t \in \mathbb{R}^{d_4 \times d_1}$, $b_t \in \mathbb{R}^{d_1}$, $W_e \in \mathbb{R}^{d_4 \times d_3}$, $b_e \in \mathbb{R}^{d_3}$ are parameters to be learned. $\hat{h}'_i$ is the mixed fusion embedding. $\hat{h}'_{e_j}$ and $\hat{h}'_{t_i}$ are regenerated entity and textual embeddings, respectively.

### 3.2 Multi-Level Knowledge-aware Augmenter

It enables the model to learn more fine-grained semantic gaps of injected knowledge triplets, leveraging the locally dense characteristics to further remedy the global sparsity problem. We focus on constructing positive and negative samples of

higher quality with multiple difficulty levels via the point-biconnected components subgraph structure. In this section, we focus on the sample construction process shown in Figure 4. The training task of this module is introduced in Sec. 3.3.

### 3.2.1 Positive Sample Construction

We extract $\mathcal{K}$ neighbor triples of the target entity $e_0$ as positive samples, which are closest to the target entity in the neighboring candidate subgraph structure. The semantic information contained in these triples is beneficial to enhancing contextual knowledge. To better aggregate target entity and contextual tokens representations, $\mathcal{K}$ neighbor triples are concatenated together into a sentence. We obtain the unified semantic representation via a shared *Text Encoder* (e.g., BERT (Devlin et al., 2019)). Since the semantic discontinuity between the sampling of different triples from discrete entities and relations, we modify the position embeddings such that tokens of the same triple share the same positional index, and vice versa. For example, the position of the input tokens in Fig. 4 triple $(e_0, r(e_0, e_1), e_1)$ is all 1. To unify the representation space, we take the [CLS] (i.e., the first token of input format in the BERT) representation as positive sample embedding to represent sample sequence information. We formulate $h_{p_j} \in \mathbb{R}^{d_1}$ as the positive embedding of an entity word $t_{je}^C$.

### 3.2.2 Point-biconnected Component-based Negative Sample Construction

In closed-domain KGs, nodes are densely connected to the neighbouring nodes owning to the locally dense property which is conducive to graph searching. Therefore, we search for a large amount of nodes that are further away from target entities as negative samples. For example in Figure 4, we construct a negative sample by the following steps:

- **STEP 1:** Taking the starting node $e_{start}$ (i.e. $e_0$) as the center point and searching outward along the relations, we obtain end nodes $e_{end}$ with different hop distance $\text{Hop}(P(\mathcal{G}, e_{start}, e_{end}))$ where $\text{Hop}(\cdot)$ denotes the hop distance and $P(\mathcal{G}, e_i, e_j)$ denotes the shortest path between $e_i$ and $e_j$ in the graph $\mathcal{G}$. For example, $\text{Hop}(P(\mathcal{G}, e_0, e_{10})) = 2$ in Path 3 and $\text{Hop}(P(\mathcal{G}, e_0, e_{11})) = 3$ in Path 6.

- **STEP 2:** We leverage the hop distance to construct negative samples with different structurally difficulty levels, where $\text{Hop}(\cdot) = 2$ for

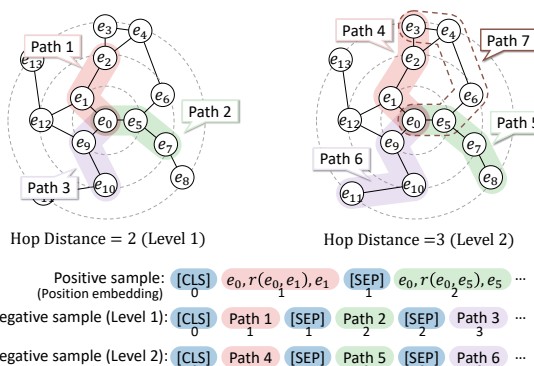

Figure 4: Examples of positive and negative sample construction. We add the [SEP] token between paths to differentiate triplets. Note that the subscripts are positional embedding indexes.

Level-1 and $\text{Hop}(\cdot) = $ n+1 for Level-$n$ samples. We assume that the closer the hop distance is, the more difficult it is to distinguish the semantic knowledge contained between triples w.r.t. the starting node.

- **STEP 3:** The constructed pattern of negative samples is similar to positive samples whose paths with the same distance are merged into sentences. Note that we attempt to choose the shortest path (e.g., Path 4) when nodes' pairs contain at least two disjoint paths (i.e., point-biconnected component). For each entity, we build negative samples of $k$ levels.

For the high proportions of Point-biconnected Component in closed-domain KGs, there are multiple disjoint paths between starting nodes and end nodes in most cases such as Path 4 and Path 7 in Figure 4. We expand the data augmentation strategy that prefers end node pairs with multiple paths and adds the paths to the same negative sample, enhancing sample quality with diverse information. The relationships among these node pairs contain richer and indistinguishable semantic information. Besides, our framework preferentially selects nodes in the same entity class of target entity to enhance the difficulty and quality of samples. Negative sample embeddings are formulated as $\{h_{n_j}^{(l)}\}_{l=1}^k$ where $h_{n_j}^{(l)} \in \mathbb{R}^{d_1}$ and $l$ present various different level of negative samples. The specific algorithm description of the negative sample construction process is shown in Appendix Algorithm 1.

| Models ↓   Tasks → | Financial | | | | | | Medical | | | |
|---|---|---|---|---|---|---|---|---|---|---|
| | NER | TC | QA | QM | NED | Average | NER | QNLI | QM | Average |
| RoBERTa | 78.92 | 82.61 | 82.20 | 91.28 | 92.56 | $85.51_{\pm 0.53}$ | 65.89 | 94.63 | 85.68 | $82.07_{\pm 0.31}$ |
| BERT | 77.56 | 83.68 | 83.01 | 91.70 | 92.46 | $85.68_{\pm 0.28}$ | 70.24 | 94.60 | 84.82 | $83.22_{\pm 0.23}$ |
| $Con_{pt}$ | 80.67 | 84.43 | 83.98 | 91.96 | 92.78 | $86.76_{\pm 0.19}$ | 73.35 | 94.46 | 86.38 | $84.73_{\pm 0.22}$ |
| ERNIE-Baidu | 82.99 | 84.75 | 84.46 | 92.26 | 92.39 | $87.37_{\pm 0.14}$ | 75.60 | 95.24 | 86.02 | $85.62_{\pm 0.20}$ |
| ERNIE-THU | 87.19 | 85.35 | 83.83 | 92.51 | 92.86 | $88.35_{\pm 0.17}$ | 80.36 | 95.85 | 86.42 | $87.54_{\pm 0.12}$ |
| KnowBERT | 86.89 | 85.13 | 82.74 | 92.09 | 92.33 | $87.84_{\pm 0.21}$ | 79.84 | 94.97 | 85.87 | $86.89_{\pm 0.14}$ |
| K-BERT | 86.28 | 85.41 | 84.56 | 92.41 | 92.52 | $88.24_{\pm 0.25}$ | 79.72 | 95.93 | 86.64 | $87.43_{\pm 0.19}$ |
| KGAP | 85.43 | 84.91 | 83.76 | 91.95 | 92.04 | $87.62_{\pm 0.15}$ | 77.25 | 94.88 | 86.34 | $86.16_{\pm 0.24}$ |
| DKPLM | 86.83 | 85.22 | 83.95 | 91.86 | 92.57 | $88.08_{\pm 0.17}$ | 80.11 | 94.97 | 85.88 | $86.99_{\pm 0.14}$ |
| GreaseLM | 85.62 | 84.82 | 83.94 | 91.85 | 92.26 | $87.70_{\pm 0.23}$ | 78.73 | 95.46 | 85.92 | $86.70_{\pm 0.22}$ |
| KALM | 87.06 | 85.38 | 82.96 | 91.87 | 92.64 | $87.98_{\pm 0.16}$ | 80.12 | 94.84 | 85.93 | $86.96_{\pm 0.13}$ |
| KANGAROO$^{\triangle}$ | 87.41 | 85.22 | 84.02 | 92.46 | 92.96 | $88.41_{\pm 0.16}$ | 80.57 | 95.32 | 86.19 | $87.36_{\pm 0.20}$ |
| KANGAROO$^{\ddagger}$ | **88.16** | **86.58** | **84.92** | **93.26** | **93.54** | $\mathbf{89.29_{\pm 0.13}}$ | **81.19** | **96.15** | **87.42** | $\mathbf{88.25_{\pm 0.17}}$ |

Table 2: The performance of fully-supervised learning in terms of F1 (%). $^{\triangle}$ and $^{\ddagger}$ indicate that we pre-train our model on CN-DBpedia (i.e., open domain KG) and the corresponding close-domain KG, respectively. The results of knowledge-aware PLMs baselines are pre-trained in closed-domain KGs. Best performance is shown in **bold**.

## 3.3 Training Objectives

In our framework, the training objectives mainly consist of two parts, including the masked language model loss $\mathcal{L}_{MLM}$ (Devlin et al., 2019) and the contrastive learning loss $\mathcal{L}_{CL}$, formulated as follows:

$$\mathcal{L}_{total} = \lambda_1 \mathcal{L}_{MLM} + \lambda_2 \mathcal{L}_{CL} \qquad (8)$$

where $\lambda_1$ and $\lambda_2$ are the hyperparameters. As for the multi-level knowledge-aware contrastive learning loss, we have obtained the positive sample $\hat{h}'_{e_j}$ and negative samples $h^{(l)}_{n_j}$ for each entity target $\hat{h}'_{t_j}$ (i.e. the textual embedding of token in an entity word). We take the standard InfoNCE (van den Oord et al., 2018) as our loss function $\mathcal{L}_{CL}$:

$$\mathcal{L}_{CL} = -\sum_j \log \frac{e^{\cos(\hat{h}'_{t_j}, \hat{h}'_{e_j})/\tau}}{e^{\cos(\hat{h}'_{t_j}, \hat{h}'_{e_j})/\tau} + \sum_l e^{\cos(\hat{h}'_{t_j}, h^{(l)}_{n_j})/\tau}} \qquad (9)$$

where $\tau$ is a temperature hyperparameter and $cos$ is the cosine similarity function.

## 4 Experiments

In this section, we conduct extensive experiments to evaluate the effectiveness of the proposed framework. Due to the space limitation, the details of datasets and model settings are shown in Appendix B and the baselines are described in Appendix C.

### 4.1 Results of Downstream Tasks

**Fully-Supervised Learning** We evaluate the model performance on downstream tasks which are shown in Table 2. Note that the input format

of NER task in financial and medical domains are related to knowledge entities and the rest are implicitly contained. The fine-tuning models use a similar structure compared to KANGAROO, which simply adds a linear classifier at the top of the backbone. From the results, we can observe that: (1) Compared with PLMs trained on open-domain corpora, KEPLMs with domain corpora and KGs achieve better results, especially for NER. It verifies that injecting the domain knowledge can improve the results greatly. (2) ERNIE-THU and K-BERT achieve the best results among baselines and ERNIE-THU performs better in NER. We conjecture that it benefits from the ingenious knowledge injection paradigm of ERNIE-THU, which makes the model learn rich semantic knowledge in triples. (3) KANGAROO greatly outperforms the strong baselines improves the performance consistently, especially in two NER datasets (+0.97%, +0.83%) and TC (+1.17%). It confirms that our model effectively utilizes the closed-domain KGs to enhance structural and semantic information.

**Few-Shot Learning** To construct few-shot data, we sample 32 data instances from each training set and employ the same dev and test sets. We also fine-tune all the baseline models and ours using the same approach as previously. From Table 3, we observe that: (1) The model performance has a sharp decrease compared to the full data experiments. The model can be more difficult to fit testing samples by the limited size of training data. In general, our model performs best in all the baseline results. (2) Although ERNIE-THU gets the best score in Question Answer, its performances

on other datasets are far below our model. The performance of KANGAROO, ERNIE-THU and K-BERT is better than others. We attribute this to their direct injection of external knowledge into textual representations.

## 4.2 Detailed Analysis of KANGAROO

### 4.2.1 Ablation Study

We conduct essential ablation studies on four important components with Financial NER and Medical QM tasks. The simple triplet method simplifies the negative sample construction process by randomly selecting triplets unrelated to target entities. The other three ablation methods respectively detach the entity-class embeddings, the contrastive loss and the masked language model (MLM) loss from the model and are re-trained in a consistent manner with KANGAROO. As shown in Table 4, we have the following observations: (1) Compared to the simple triplet method, our model has a significant improvement. It confirms that Point-biconnected Component Data Augmenter builds rich negative sample structures and helps models learn subtle structural semantic to further compensate the global sparsity problem. (2) It verifies that entity class embeddings and multi-level contrastive learning pre-training task effectively complement semantic information and make large contributions to the complete model. Nonetheless, without the modules, the model is still comparable to the best baselines ERNIE-THU and K-BERT.

### 4.2.2 The Influence of Hyperbolic Embeddings

In this section, we comprehensively analyze why hyperbolic embeddings are better than Euclidean embeddings for the closed-domain entity-class hierarchical structure.

**Visualization of Embedding Space**. To compare the quality of features in Euclidean and hyperbolic spaces, we train KG representations by TransE (Bordes et al., 2013) and the Poincaré ball model (Nickel and Kiela, 2017), visualizing the embeddings distribution using t-SNE dimensional reduction (van der Maaten and Hinton, 2008) shown in Figure 5. They both reflect embeddings grouped by classes, which are marked by different colors. However, TransE embeddings are more chaotic, whose colors of points overlap and hardly have clear boundaries. In contrast, hyperbolic representations reveal a clear hierarchical structure. The root node is approximately in the center and links

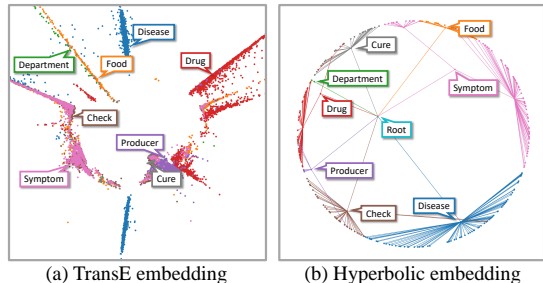

(a) TransE embedding    (b) Hyperbolic embedding

Figure 5: Visualization of TransE and hyperbolic embeddings.

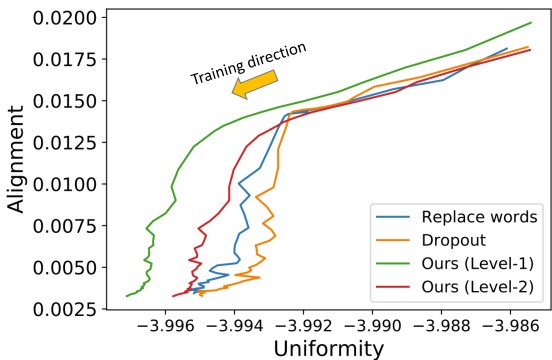

Figure 6: Results comparison of ours and other data augmentation methods of alignment and uniformity.

to the concept-level nodes such as drug, check and cure. It illustrates that hyperbolic embeddings fit the hierarchical data better and easily capture the differences between classes.

**Performance Comparison of Different Embeddings**. We replace entity-class embeddings with Euclidean embeddings to verify the improvement of the hyperbolic space. To obtain entity-class embeddings in the Euclidean space, we obtain embeddings of closed-domain KGs by TransE (Bordes et al., 2013) and take them as a substitution of entity-class embeddings. As shown in Table 5, we evaluate the Euclidean model in four downstream tasks, including NER and TC task in the financial domain, together with NER and QM in the medical domain. The results show that the performance degradation is clear in all tasks with Euclidean entity-class embeddings. Overall, the experimental results confirm that the closed-domain data distribution fits the hyperbolic space better and helps learn better representations that capture semantic and structural information.

| Models ↓   Tasks → | Financial | | | | | | Medical | | | |
|---|---|---|---|---|---|---|---|---|---|---|
| | NER | TC | QA | QM | NED | Average | NER | QNLI | QM | Average |
| RoBERTa | 69.31 | 70.95 | 71.23 | 82.44 | 83.39 | $75.64_{\pm2.07}$ | 59.87 | 61.48 | 59.82 | $60.39_{\pm1.84}$ |
| BERT | 68.46 | 72.62 | 72.81 | 81.57 | 83.61 | $75.81_{\pm1.95}$ | 63.28 | 51.72 | 57.96 | $57.65_{\pm2.58}$ |
| $Con_{pt}$ | 71.62 | 70.95 | 77.67 | 80.24 | 85.83 | $77.26_{\pm1.84}$ | 65.87 | 64.04 | 60.34 | $63.42_{\pm2.02}$ |
| ERNIE-Baidu | 75.86 | 76.85 | 76.67 | 84.69 | 85.13 | $79.84_{\pm1.83}$ | 69.44 | 64.78 | 61.74 | $65.32_{\pm2.25}$ |
| ERNIE-THU | 77.72 | 78.12 | 79.36 | 83.26 | 82.89 | $80.27_{\pm1.86}$ | 71.45 | 66.65 | 66.18 | $68.09_{\pm1.62}$ |
| KnowBERT | 79.35 | 77.62 | 79.54 | 85.77 | 84.33 | $81.32_{\pm2.13}$ | 68.84 | 65.62 | 63.96 | $66.14_{\pm1.72}$ |
| K-BERT | 76.92 | 75.11 | 78.59 | 84.87 | 83.99 | $79.90_{\pm1.70}$ | 69.81 | 65.73 | 72.60 | $69.38_{\pm1.93}$ |
| KGAP | 78.51 | 76.78 | 77.72 | 83.04 | 84.15 | $80.04_{\pm1.55}$ | 70.52 | 67.05 | 70.64 | $69.40_{\pm2.13}$ |
| DKPLM | 77.40 | 76.34 | 78.48 | 85.16 | 83.62 | $80.20_{\pm1.66}$ | 70.62 | 68.58 | 68.93 | $69.38_{\pm2.36}$ |
| GreaseLM | 79.70 | 78.24 | 77.94 | 83.06 | 84.27 | $80.64_{\pm1.92}$ | 71.47 | 67.58 | 69.41 | $69.49_{\pm2.25}$ |
| KALM | 76.93 | 76.81 | 78.03 | 84.30 | 83.84 | $79.98_{\pm1.86}$ | 70.23 | 66.87 | 70.28 | $69.13_{\pm1.94}$ |
| KANGAROO$^{\triangle}$ | 79.25 | 78.41 | 78.29 | 83.62 | 85.45 | $81.00_{\pm1.56}$ | 68.50 | 68.29 | 71.23 | $69.34_{\pm1.42}$ |
| KANGAROO$^{\ddagger}$ | **81.61** | **80.73** | **79.98** | **87.92** | **86.12** | $\mathbf{83.27_{\pm1.62}}$ | **73.42** | **70.34** | **75.32** | $\mathbf{73.03_{\pm1.72}}$ |

Table 3: The overall results of few-shot learning in terms of F1 (%).

| Models ↓ Tasks → | Fin. NER | Med. QM |
|---|---|---|
| KANGAROO$^{\ddagger}$ | **88.16** | **87.42** |
| w/o Simple Triplets | 87.81 | 86.46 |
| w/o Entity Class | 87.89 | 86.15 |
| w/o Contrastive Loss | 87.86 | 86.61 |
| w/o MLM | 87.92 | 86.78 |

Table 4: The performance of models for ablation study in terms of F1 (%).

| Tasks → | Financial | | Medical | |
|---|---|---|---|---|
| Models ↓ | NER | TC | NER | QM |
| Euclidean | 87.69 | 86.16 | 80.34 | 86.30 |
| Hyperbolic | **88.16** | **86.58** | **81.19** | **87.42** |

Table 5: The model results when Euclidean and hyperbolic embeddings are employed in terms of F1 (%).

| | Pos. | Neg. L1 / L2 / L3 |
|---|---|---|
| KANGAROO$^{\ddagger}$ | 0.0911 | 0.0061 / 0.0043 / 0.0018 |
| Dropout | 0.0991 | -0.0177 |
| Word Rep. | 0.0992 | -0.0144 |

Table 6: The averaged cosine similarities of positive/negative samples. "L1" means the Level-1 samples.

### 4.2.3 The Influence of Point Biconnected Component-based Data Augmentation

To further confirm that our data augmentation technique for contrastive learning is effective, we analyze the correlation between positive and negative samples w.r.t. target entities. We choose two strategies (i.e., dropout (Gao et al., 2021) and word replacement (Wei and Zou, 2019) for positive samples) as baselines. The negative samples are randomly selected from other entities. As shown in Table 6, we calculate the averaged cosine similarity between samples and target entities. In positive samples, the cosine similarity of our model is lower than in baselines, illustrating the diversity between positive samples and target entities. As for negative samples, we design the multi-level sampling strategy in our model, in which Level-1 is the most difficult followed by Level-2 and Level-3. The diversity and difficulty of the negative sam-

ples help to improve the quality of data augmentation. We visualize the alignment and uniformity metrics (Wang and Isola, 2020) of models during training. To make this more intuitive, we use the cosine distance to calculate the similarity between representations. The lower alignment shows similarity between positive pairs features and lower uniformity reveals presentation preserves more information diversity. As shown in Figure 6, our models greatly improve uniformity and alignment steadily to the best point.

## 5   Related Work

**Open-domain KEPLMs.** We summarize previous KEPLMs grouped into four types: (1) Knowledge enhancement by entity embeddings (Zhang et al., 2019; Su et al., 2021; Wu et al., 2023). (2) Knowledge-enhancement by text descriptions (Wang et al., 2021; Zhang et al., 2021a). (3) Knowledge-enhancement by converted triplet's texts (Liu et al., 2020a; Sun et al., 2020). (4) Knowledge-enhancement by retrieve the external text and token embedding databases (Guu et al., 2020; Borgeaud et al., 2022).

**Closed-domain KEPLMs.** Due to the lack of in-domain data and the unique distributions of domain-specific KGs (Cheng et al., 2015; Savnik

et al., 2021), previous works of closed-domain KE-PLMs focus on three domain-specific pre-training paradigms. (1) Pre-training from Scratch. For example, PubMedBERT (Gu et al., 2022) derives the domain vocabulary and conducts pre-training using solely in-domain texts, alleviating the problem of out-of-vocabulary and perplexing domain terms. (2) Continue Pre-training. These works (Beltagy et al., 2019; Lee et al., 2020) have shown that using in-domain texts can provide additional gains over plain PLMs. (3) Mixed-Domain Pre-training (Liu et al., 2020b; Zhang et al., 2021b). In this approach, out-domain texts are still helpful and typically initialize domain-specific pre-training with a general-domain language model and inherit its vocabulary. Although these works inject knowledge triples into PLMs, they pay little attention to the in-depth characteristics of closed-domain KGs (Cheng et al., 2015; Kazemi and Poole, 2018; Vashishth et al., 2020), which is the major focus of our work.

## 6 Conclusion

In this paper, we propose a unified closed-domain framework named KANGAROO to learn knowledge-aware representations via implicit KGs structure. We utilize entity enrichment with hyperbolic embeddings aggregator to supplement the semantic information of target entities and tackle the semantic deficiency caused by global sparsity. Additionally, we construct high-quality negative samples of knowledge triples by data augmentation via local dense graph connections to better capture the subtle differences among similar triples.

## Limitations

KANGAROO only captures the global sparsity structure in closed-domain KG with two knowledge graph embedding methods, including euclidean (e.g. transE (Bordes et al., 2013)) and hyperbolic embedding. Besides, our model explores two representative closed domains (i.e. medical and financial), and hence we might omit other niche domains with unique data distribution.

## Ethical Considerations

Our contribution in this work is fully methodological, namely a new pre-training framework of closed-domain KEPLMs, achieving the performance improvement of downstream tasks. Hence, there is no explicit negative social influences in this work. However, Transformer-based models may have some negative impacts, such as gender and social bias. Our work would unavoidably suffer from these issues. We suggest that users should carefully address potential risks when the KANGAROO models are deployed online.

## Acknowledgements

This work was supported by the National Natural Science Foundation of China under grant number 62202170 and Alibaba Group through the Alibaba Innovation Research Program.

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

## A  Indicators of Closed-domain KGs

The following explanations are the 5 different indicators to analyze the closed-domain KGs.

- *#Nodes* and *#Edges* are the numbers of nodes and edges in the corresponding KG.

- *Coverage Ratio* is the entity coverage rate of the KG in its corresponding text corpus. We calculate it by the percentage of entity tokens matched in the KG by the number of the full-text tokens, formulated as $CR = \frac{t_e}{t_t}$, where $t_e$ is the number of entity tokens and $t_t$ is the number of all textual tokens. Texts for closed domains are the same as the pre-training corpora to be described in the experiments. The corpora for the open domain is taken from the CLUE benchmark[5].

- *%Max Point Biconnected Component* (i.e., %Max PBC) is the number of nodes in the biggest point biconnected component divided by the number of all nodes. Note that a point biconnected component is a graph such that if any node is removed, the connectivity is not changed.

- *Subgraph Density*[6] is the average density of 100 random subgraphs where each of them contains 10% of total nodes of the KG. Note that the density of a graph $G = (V, E)$ is formulated as $\frac{|E|}{|V|(|V|-1)}$ where $|E|$ is the number of edges and $|V|$ is the number of nodes.

## B  Data and Model Settings

### B.1  Pre-training Corpus

Our experiments are conducted in two representative closed domains, including finance and medical. The pre-training corpus of the financial domain is crawled from several leading financial news websites in China, including SOHU Financial[7], Cai Lian Press[8] and Sina Finance[9], etc. The corpus of the medical domain is from a Chinese network community of professional doctors called DXY Bulletin Board System[10]. After pre-processing, the

---

[5]We use the Chinese Wikipedia corpus "wiki2019zh" and the news corpus "news2016zh" as the pre-training corpora. https://github.com/brightmart/nlp_chinese_corpus

[6]https://en.wikipedia.org/wiki/Dense_subgraph

[7]https://business.sohu.com/

[8]https://www.cls.cn/

[9]https://finance.sina.com.cn/

[10]https://www.dxy.cn/bbs/newweb/pc/home

| Dataset | | #Train | #Dev | #Test |
|---|---|---|---|---|
| | NER | 23,675 | 2,910 | 3,066 |
| | TC | 37,094 | 4,122 | 4,580 |
| Fin. | QA | 612,466 | 76,781 | 76,357 |
| | QM | 81,000 | 9,000 | 10,000 |
| | NED | 4,049 | 450 | 500 |
| | NER | 34,209 | 8,553 | 8,553 |
| Med. | QM | 16,067 | 1,789 | 1,935 |
| | QNLI | 80,950 | 9,066 | 9,969 |

Table 7: The number of samples of the datasets in financial and medical domain, respectively.

financial corpus contains 12,321,760 text segments and that number of segments in the medical corpus is 9,504,007.

## B.2 Knowledge Graph

As for KGs, FinKG contains five types of classes including companies, industries, products, people and positions with 9,413 entities and 18,175 triples. The MedKG is disease-related, containing diseases, drugs, symptoms, cures and pharmaceutical factories. It consists of 43,972 entities and 296,625 triples.

## B.3 Downstream Tasks

We use five financial datasets and three medical datasets to evaluate our model in full and few-shot learning settings. Financial task data is obtained from public competitions and previous works, including Named Entity Recognition[11] (NER), Text Classification[12] (TC), Question Answering[13] (QA), Question Matching[14] (QM) and Negative Entity Discrimination[15] (NED). Medical data is taken from ChineseBlue[16] tasks and DXY Company[17]. Medical datasets contain NER, Question Matching and Question Natural Language Inference (QNLI). For most competitions where only train and dev sets are available for us, we randomly slice the datasets, making the proportion of train/dev/test ratio close to 9/1/1. The train/dev/test data distribution and data sources are shown in Table 7.

[11] https://embedding.github.io/
[12] https://www.biendata.xyz/competition/ccks_2020_4_1/data/
[13] https://github.com/autoliuweijie/K-BERT
[14] https://www.biendata.xyz/competition/CCKS2018_3/
[15] https://www.datafountain.cn/competitions/353/datasets
[16] https://github.com/alibaba-research/ChineseBLUE
[17] https://auth.dxy.cn

**Algorithm 1** Negative Samples Construction

1: **Input:** Knowledge graph $\mathcal{G} = (\mathcal{E}, \mathcal{R})$, entity $e_{\text{start}}$, level $D$, sample length $L$
2: **Output:** Sequence of negative sample $S$
3: $S \leftarrow$ empty sequence
4: **while** $\text{len}(S) < L$ **do**
5: $\quad N(e_{\text{start}}) \leftarrow \{e \in \mathcal{E} | \text{Hop}(P(\mathcal{G}, e_{\text{start}}, e)) = D + 1\}$
6: $\quad N_s(e_{\text{start}}) \leftarrow \{e \in N(e_{\text{start}}) | \text{class}(e_{\text{start}}) = \text{class}(e)\}$
7: $\quad$ **if** $N_s(e_{\text{start}}) = \varnothing$ **then**
8: $\quad\quad e_{\text{end}} \leftarrow \text{random\_select}(N(e_{\text{start}}))$
9: $\quad$ **else**
10: $\quad\quad e_{\text{end}} \leftarrow \text{random\_select}(N_s(e_{\text{start}}))$
11: $\quad$ **end if**
12: $\quad S \leftarrow \text{concatenate}(S, P(\mathcal{G}, e_{\text{start}}, e_{\text{end}}))$
13: $\quad \mathcal{E}' \leftarrow \mathcal{E} - P(\mathcal{G}, e_{\text{start}}, e_{\text{end}}) \cup \{e_{\text{start}}, e_{\text{end}}\}$
14: $\quad \mathcal{R}' \leftarrow \{r(e_1, e_2) | e_1, e_2 \in \mathcal{E}'\}$
15: $\quad \mathcal{G}' \leftarrow (\mathcal{E}', \mathcal{R}')$
16: $\quad$ **if** $e_{\text{start}}$ and $e_{\text{end}}$ are connected in $\mathcal{G}'$ **then**
17: $\quad\quad S \leftarrow \text{concatenate}(S, P(\mathcal{G}', e_{\text{start}}, e_{\text{end}}))$
18: $\quad$ **end if**
19: **end while**
20: **return** $S$

## B.4 Model Hyperparameters

We adopt the parameters of BERT released by Google[18] to initialize the Transformer blocks for text encoding. For optimization, we set the learning rate as $5e^{-5}$, the max sequence length as 128, and the batch size as 512. The dimension of embedding $d_1$, $d_2$, $d_3$ and $d_4$ are set as 768, 100, 100 and 768, respectively. The number of the *Text Encoder* layers $N$ is 5 and the number of the *Knowledge Encoder* layers $M$ is 6. The temperature hyperparameter $\tau$ is set to 1 and the coefficients $\lambda_1$ and $\lambda_2$ are both $0.5$. The number of negative samples levels $k$ is 3. Pre-training KANGAROO takes about 48 hours per epoch on 8 NVIDIA A100 GPUs (80GB memory per card). Results are presented in average with 5 random runs with different random seeds and the same hyperparameters.

## B.5 Model Notations

We denote the input token sequence as $\{t_1, t_2, ..., t_{je}^C, ...t_n\}$ where $n$ is the length of input sequence. $t_{je}^C$ means one of the tokens of an entity word that belongs to a specific entity class $C$, such as the disease class in the medical domain. We obtain hidden feature of tokens $\{h_{t_i}\}_{i=1}^n \in \mathbb{R}^{d_1}$ by the *Text Encoder* which is composed by $N$ Transformer layers (Vaswani et al., 2017) where $d_1$ is the PLM's output dimension. $d_2$ is the dimension of entity class embedding. Furthermore, the knowledge graph $\mathcal{G} = (\mathcal{E}, \mathcal{R})$ consists of the entities set $\mathcal{E}$ and the relations set $\mathcal{R}$. The

[18] https://github.com/google-research/bert

| Models ↓  Tasks → | Music | | Social | Average |
|---|---|---|---|---|
| | D1 | D2 | D1 | - |
| RoBERTa | 57.34 | 70.86 | 63.29 | $63.83_{\pm 0.18}$ |
| BERT | 54.26 | 67.13 | 62.58 | $61.32_{\pm 0.21}$ |
| $Con_{pt}$ | 59.43 | 72.04 | 63.97 | $65.15_{\pm 0.20}$ |
| ERNIE-Baidu | 59.43 | 73.60 | 65.88 | $66.30_{\pm 0.14}$ |
| ERNIE-THU | 58.62 | 72.92 | 65.10 | $65.55_{\pm 0.17}$ |
| KnowBERT | 59.91 | 74.84 | 66.36 | $67.04_{\pm 0.24}$ |
| K-BERT | 58.74 | 73.71 | 67.01 | $66.49_{\pm 0.19}$ |
| KGAP | 59.62 | 74.35 | 66.82 | $66.93_{\pm 0.22}$ |
| DKPLM | 57.33 | 73.64 | 67.43 | $66.13_{\pm 0.15}$ |
| GreaseLM | 60.18 | 74.95 | 68.14 | $67.76_{\pm 0.20}$ |
| KALM | 58.94 | 73.55 | 67.85 | $66.78_{\pm 0.15}$ |
| KANGAROO[‡] | **61.51** | **75.89** | **69.27** | $68.89_{\pm 0.18}$ |

Table 8: Results of other domains in terms of Acc (%).

triplet set is $S_t = \{(e_i, r(e_i, e_j), e_j) \mid e_i, e_j \in \mathcal{E}, r(e_i, e_j) \in \mathcal{R}\}$ where $e_i$ is the head entity with relation $r(e_i, e_j)$ to the tail entity $e_j$.

## C  Baselines

In our experiments, we compare our model with general PLMs and KEPLMs in base level parameters with knowledge embeddings injected: **General PLMs:** *BERT* (Devlin et al., 2019) is a pre-trained Transformer layers initialized by public weights. *RoBERTa* (Liu et al., 2019) is the RoBERTa model pre-trained with a Chinese corpus, which improves dynamic masking and the training strategy of BERT. $Con_{pt}$ is the continual pre-trained BERT on domain pre-training data. It further helps to improve the original BERT model performance in closed domains.

**KEPLMs:** For the fairness of the results, all the KEPLMs are reproduced via the closed domain KG and our pre-training corpus using official open source code. *ERNIE-Baidu* is the KEPLM (Sun et al., 2019) that adds external knowledge through entity and phrase masking. *ERNIE-THU* (Zhang et al., 2019) encodes the graph structure of KGs with knowledge embedding algorithms and injects it into contextual representations. *KnowBERT* (Peters et al., 2019) proposes the entity linkers and self-supervised language modeling objective are jointly trained end-to-end in a multitask setting that combines a small amount of entity linking supervision with a large amount of raw text. *K-BERT* (Liu et al., 2020a) is a KEPLM that converts the triples into the sentences as domain knowledge. *KGAP* (Feng et al., 2021) adopts relational graph neural networks and conduct political perspective detection as graph-level classification tasks. *DKPLM* (Zhang

et al., 2021c) specifically focuses on long-tail entities, decomposing the knowledge injection process of three PLMs' stages including pre-training, fine-tuning and inference. *GreaseLM* (Zhang et al., 2022b) fuses encoded representations from PLMs and graph neural networks over multiple layers of modality interaction operations. *KALM* (Feng et al., 2022) jointly leverages knowledge in local, document-level, and global contexts for long document understanding.

## D  Model Performance

### D.1  Other Domain Results

In the table 8, we supplement data from two domains including music domain and social domain to further demonstrate the effectiveness of the proposed KANGAROO model. We conduct experiments in the largest Chinese knowledge graph database (i.e., openKG). Both the music and social data are crawled from the largest open-source knowledge graph database in Chinese and Baidu BaiKe. We evaluate the baselines and our kangaroo model in full data fine-tuning settings and the results are shown are follows. The downstream datasets of music D1, music D2 and social D1 are text classification tasks, which 'D1' means Dataset 1. Specifically, Music D1 and D2 are the music emotion text classification tasks that download song comment data on the music app. Social D1 is classifying the relationships between public figures such as history and entertainment. The evaluation metric is accuracy (i.e., ACC).

### D.2  K-hop Thresholds' Results

In the table 9, we consider three different situations to discuss the k-hop thresholds selection including (1) k-hop triple path as positive and k+1, k+2, k+3 hop triple path as negative (2) k, k+1, k+2 triple path as negative and k+3 triple path as positive (3) k hop triple path viewed as both positive and negative samples and k+1, k+2 as negative samples. Specifically, we select the original closed domain pre-training and KG data to perform full data fine-tuning tasks. For the above three situations, in order to prevent overlapping triple path's conflicts between positive and negative during the sampling process, we mask the sampled triple path data in the iterative sampling process.

The "S1" means "Situations 1". Fin and Med means Financial and Medical respectively. From the above table, we can observe that (1) The closer

| Situations | Models | Fin NER | Fin TC | Fin QA | Fin QM | Fin NED | Med NER | Med QNLI | Med QM |
|---|---|---|---|---|---|---|---|---|---|
| S1 | k=1 | 88.16 | 86.58 | 84.92 | 93.26 | 93.54 | 81.19 | 95.15 | 87.42 |
| S1 | k=1 | 87.43 | 85.73 | 84.06 | 92.50 | 92.78 | 80.31 | 95.98 | 86.48 |
| S1 | k=3 | 85.33 | 83.27 | 81.97 | 91.02 | 91.42 | 76.64 | 93.75 | 84.81 |
| S2 | k=1 | 82.08 | 82.94 | 81.74 | 90.27 | 90.98 | 70.38 | 93.28 | 84.06 |
| S2 | k=2 | 80.96 | 82.28 | 80.85 | 89.85 | 88.33 | 68.85 | 90.44 | 82.15 |
| S2 | k=3 | 79.45 | 79.96 | 78.49 | 87.46 | 87.56 | 64.72 | 87.30 | 81.57 |
| S3 | k=1 | 87.82 | 86.26 | 84.63 | 92.98 | 93.18 | 80.90 | 95.03 | 87.20 |
| S3 | k=2 | 85.29 | 83.55 | 82.50 | 91.54 | 91.82 | 78.75 | 93.27 | 85.32 |
| S3 | k=3 | 81.99 | 81.30 | 80.41 | 90.66 | 88.46 | 75.41 | 90.73 | 81.25 |

Table 9: Results w.r.t. k-hop thresholds.

the positive sample hop path is to the target entity, the better the model performance (See S1 results). (2) The closer the negative samples are sampled to the target entity or even closer than positive samples, the performance of the model will sharply decrease (See S2 and S3 results). Hence, we should construct positive samples closer to the target entity, while negative samples should not be too far away simultaneously in graph path to avoid introducing too much knowledge noise.