# OpenReview forum: "Learning Knowledge-Enhanced Contextual Language Representations for Domain Natural Language Understanding"
_EMNLP/2023/Conference — EMNLP 2023 Main_

### Official Review · Reviewer_fcFt · 2023-07-21

**Soundness:** 3

**Excitement:**

4: Strong: This paper deepens the understanding of some phenomenon or lowers the barriers to an existing research direction.

**Paper Topic And Main Contributions:**

The paper proposed a model for injecting domain knowledge on a pretrain language model. The main contribution includes the use of hyperbolic embeddings and a robust negative sampling to address local density problem.

**Questions For The Authors:**

What is the running time of the negative sampling methods?

**Reasons To Accept:**

1. The paper is well written. The proposed model is well-justified and clearly explained.
2. The proposed model achieves state-of-the-art results.

**Reasons To Reject:**

1. Lack of experiments in general domain KG injection into pre-trained language models.

**Reproducibility:**

3: Could reproduce the results with some difficulty. The settings of parameters are underspecified or subjectively determined; the training/evaluation data are not widely available.

**Reviewer Confidence:**

4: Quite sure. I tried to check the important points carefully. It's unlikely, though conceivable, that I missed something that should affect my ratings.

---

> ### Author Rebuttal · Authors · 2023-08-28
>
> # Lack of experiments in general domain KG injection into pre-trained language models
> （1）In this domain knowledge enhanced pre-trained language scenarios, our main experiments are evaluated our model in domain data setting including pre-training data and knowledge graph data. In order to verify our model in general domain, we also supplement the experimental results in Table 2 KANGAROO$^\Delta$ settings. From the experimental results, it can be seen that our kangaroo algorithm is more effective for domain models than general data enhanced model.
> （2）Due to the limited time available for rebuttal stage and the large amount of data in the general domain, we will further supplement the relevant data results after paper acceptance. At present, we can only achieve these data results during the rebuttal stage, selecting three recent SOTA KEPLMs (i.e., GreaseLM, KALM and DKPLM). All the Chinese open domain pre-training data is constructed by authoritative institution WUDAO and the open domain KG data is the same as KANGAROO$^\Delta$ downloaded from the CN-DBpedia. The experimental results are full data fine-tuning settings.
>
> | Model |  Fin NER | Fin TC | Fin QA | Fin QM| Fin NED| Med NER | Med QNLI| Med QM|
> | :--------: | :--------: | :--------: | :--------: | :--------: |  :--------: | :--------: | :--------: | :--------: |
> | DKPLM     | 83.25 | 81.64 | 79.83 | 90.34 | 92.59 | 68.71 | 93.22 | 84.90|
> | GreaseLM | 81.86 | 83.95 | 82.86 | 90.31 | 92.62 | 74.52 | 93.80 | 84.98 |
> | KALM.       | 82.73 | 82.93 | 82.16 | 91.01 | 90.37 | 76.82 | 94.08 | 85.47 |
> | KANGAROO     | 84.99 |  83.81| 83.63 | 91.48 | 92.58 | 78.11 | 94.13 | 86.32 |
>
> From the above table, we can draw a conclusion roughly: (1) The results of knowledge augmentation method trained using large-scale pre-trained corpus in the open domain is generally better than the underlying model (i.e., BERT, ROBERTA) in the closed domain. We conjecture that the size of the pre-trained corpus in open and closed domain resulting in the KG data can not be used effectively. (2) Our model can still demonstrate superior performance in open domain large-scale pre-trained corpus.
>
> # What is the running time of the negative sampling methods?
> For the running time of the negative sampling methods, it will vary depending on the dense structure of the KG data in different domains and computing power of the machine. According to our domain KG and machine, we need about 1~2 seconds for choosing negative samples for a target entities, whereas the speed of positive triple sampling is very fast about 0.1 seconds.
>
> We sincerely thank for your time and efforts.

---

### Official Review · Reviewer_Z6FT · 2023-08-05

**Soundness:** 4

**Excitement:**

4: Strong: This paper deepens the understanding of some phenomenon or lowers the barriers to an existing research direction.

**Paper Topic And Main Contributions:**

This paper focuses on Knowledge-Enhanced Pre-trained Language Models (KEPLMs) and proposes a Knowledge-enhanced lANGuAge Representation learning framework for various clOsed dOmains (KANGAROO).

**Questions For The Authors:**

1. In Equation 5, how the multi-headed self-attention can have the multi-inputs and multi-outputs.

2. Why set only one-hop node in the knowledge graph as positive samples? And why set all one-hop nodes in the knowledge graph as positive samples?

3. Why one-hop nodes in the knowledge graph can not be negative samples?

4. Give more details in Section 4.2.3.

5. Give more details in Equation 5.

**Reasons To Accept:**

* This paper proposes one method with two components for enhancing the training process based on the structure of the close-domain Knowledge domain of Pre-trained Language Models.

* The authors evaluate their approach on two open datasets, providing a comprehensive evaluation of the proposed method. Based on the reported results, the proposed method may be effective.

**Reasons To Reject:**

The primary concerns of this paper are,

* Some report results are close to baselines. The authors should do a significate test to prove the performance and improvements are significant.

* The setting of positive and negative samples in Section 3.2.1 is quite strange. First, the motivation for utilizing all first-hop nodes in the knowledge graph as the positive sample is unclear. Second, the motivation for utilizing other hops' nodes in the knowledge graph as negative samples is also unclear. Third, the experiments didn't show if the positive and negative samples' settings changed, how the result will be changed. Fourth, for negative samples, the authors choose the path from the centre node. It must include the positive samples.

Other minor concerns include that in Section 4.2.3, the motivation and setting (such as baselines) are unclear. Specifically, I can not understand the baseline from the current description.

Then, in Equation 5, how the multi-headed self-attention can have the multi-inputs and multi-outputs.

**Reproducibility:**

4: Could mostly reproduce the results, but there may be some variation because of sample variance or minor variations in their interpretation of the protocol or method.

**Reviewer Confidence:**

3: Pretty sure, but there's a chance I missed something. Although I have a good feel for this area in general, I did not carefully check the paper's details, e.g., the math, experimental design, or novelty.

---

> ### Author Rebuttal · Authors · 2023-08-28
>
> # Some report results are close to baselines. The authors should do a significate test to prove the performance and improvements are significant
> In all experimental settings, results are presented in average with 5 random runs with different random seeds and the same hyperparameters (see Appendix B.4) to ensure the reliability and significance of the results. Meanwhile, these domain tasks are relatively difficult due to the domain text term understanding. Both our model and the previous baselines models have not made much progress, failing to achieve the significant performance improvement that was expected. To further ensure the significance of the results, we conduct a t-tests experiment demonstrate the improvements of models are statistically significant with p < 0.05 level using the original output results.
>
> # The setting of positive and negative samples in Section 3.2.1 is quite strange.
> (1) For the positive and negative samples construction, these NLP data augmentation methods in contrastive learning are generally heuristic construction based on task scenarios such as dropout, delete word and back-translation. In KEPLMs, the retrieved knowledge triples viewed as positive or negative are defined based on the distance on the graph. The closer the distance in graph, the smaller the semantic gap of these triples. Meanwhile, considering that if the negative samples are too far away in the KG, the model finds it too easy to distinguish between positive and negative samples through contrastive learning, which is not conducive to the robustness of the model.
>
> (2) For the negative sample include the positive samples problem, we select the negative triple graph path exclude the positive samples.  There is usually more than one triple around the target entity.
>
> (3) For the positive and negative samples' settings changed problems,  we discuss the changes in different negative samples and the corresponding changes in section 4.3.2 using two important metrics in contrastive learning (i.e., Alignment and Uniformity). Our learning speed is faster and more stable and we perform the specific result as follows. We evaluate our model with different negative sample methods in three medical and financial tasks with full data fine-tuning settings during the rebuttal stage.
> | Model |  Medical QM | Medical QNLI | Financial TC |
> | :--------: | :--------: | :--------: | :--------: |
> | Replace Word | 87.05 | 95.73 | 86.04 |
> | Dropout | 87.17 | 95.84 | 86.29 |
> | Our Level-1 | 87.94 | 96.11 | 86.35 |
> | Our Level-2 | 88.19 | 96.08 | 86.49 |
>
> # Give more details in Equation 5
> In Equation 5, we generally formalize the input and output of each layer of the Transformer into a function $\mathcal{F}_m^v$. In our implementation, our model not only includes multi-head self attention but also includes FFN layers, which means its inputs are the same as the Transformer inputs of each layer. Hence, our model input in Equation 5 can be viewed as Transformer input.
>
> We sincerely thank for your time and efforts.

---

### Official Review · Reviewer_m3BN · 2023-08-07

**Soundness:** 4

**Excitement:**

4: Strong: This paper deepens the understanding of some phenomenon or lowers the barriers to an existing research direction.

**Paper Topic And Main Contributions:**

The authors propose a Knowledge enhanced language representation learning framework for closed-domain tasks by capturing implicit graph structures among the entities. To achieve such a goal, the KANGAROO system exploits hyperbolic embeddings of deep hierarchical entity-class structures, avoiding entities' global sparsity and locally dense neighbour subgraphs through a data augmentation (for negative triples) strategy based on contrastive learning.

In detail, the framework contains an Entity Encoder, divided into an entity space infusion and entity knowledge injector, to encode textual input by including knowledge, semantic and structural information. While the model learns the last two data, the former is injected in the embeddings holding the semantic and structural information of the related domain. A Multi-level knowledge-aware augmenter intervenes to learn the semantic gaps of injected triples by augmenting locally existing data with negative samples. Then, the model dedicated to solving a Named Entity Recognition (NER) task learns to solve the specific goals by exploiting the principle of the masked language model and the contrastive learning paradigm.

The experiments show superior performances compared to the baseline, besides providing an in-depth ablation study to show the various components' influences.

**Reasons To Accept:**

The authors implement an exciting framework to advance the NER state-of-the-art regarding two closed domains. However, the designed solutions hold a general approach to the problem that can migrate to other closed domains.

The proposal presents a strong structure, a good formalisation of the discussed issue and an in-depth description of the implemented framework. Also, the authors made available their code to reproduce the entire experiment.

The idea is intriguing and produces evident advances in the state-of-the-art. Also, it may lead to further and stimulating discussion about the characteristics of Knowledge we are currently employing in specific scenarios. The EMNLP audience will receive benefits in discussing such a proposal.

**Reasons To Reject:**

The related work needs improvements since it is just a list of works without effectively positioning the proposal in the current literature. Its similarities and novelty are here to be shown.

Also, the limitations could be more actively written. It misses some limitations, especially technical, that the framework has in leveraging close-domain tasks.

Section 3.1 may have further details about the component and how it interacts with the remaining ones instead of barely introducing its submodules, resulting in redundancy with the previous few lines of Section 3.

Evaluating the proposed system on further domains would completely enlighten the advantages and issues that KANGAROO has in its application, revealing additional insights that may depend on the execution domain.

**Reproducibility:**

4: Could mostly reproduce the results, but there may be some variation because of sample variance or minor variations in their interpretation of the protocol or method.

**Reviewer Confidence:**

4: Quite sure. I tried to check the important points carefully. It's unlikely, though conceivable, that I missed something that should affect my ratings.

---

> ### Author Rebuttal · Authors · 2023-08-28
>
> # Section 3.1 may have further details about the component and how it interacts with the remaining ones instead of barely introducing its submodules, resulting in redundancy with the previous few lines of Section 3.
> The Hyperbolic Knowledge-aware Aggregator module in section 3.1 mainly integrates the representation of domain knowledge in hyperbolic space with the context-aware representation generated by pre-trained language models. Then the fused knowledge-enhanced token representations are transferred to pre-training tasks, performing the underlying MLM task and our proposed section 3.2 Multi-Level Knowledge-aware Contrastive Learning Augmenter. The knowledge-enhanced token representations further benefit the positive and negative samples' representations due to the point-biconnected components subgraph implicit structure.
>
> # Evaluating the proposed system on further domains would completely enlighten the advantages and issues that KANGAROO has in its application, revealing additional insights that may depend on the execution domain.
> Due to the limited time available for the rebuttal stage, we first supplement data from two domains including music domain and social domain to further demonstrate the effectiveness of the proposed KANGAROO model. We conduct experiments in the largest Chinese knowledge graph database (i.e., openKG). Both the music and social data are crawled from the largest open-source knowledge graph database in Chinese and Baidu BaiKe. We evaluate the baselines and our kangaroo model in full data fine-tuning settings and the results are shown are follows. The downstream datasets of music D1, music D2 and social D1 are text classification tasks, which 'D1' means Dataset 1. Specifically, Music D1 and D2 are the music emotion text classification tasks that download song comment data on the music app. Social D1 is classifying the relationships between public figures such as history and entertainment. The evaluation metric is accuracy (i.e., ACC). We will supplement the results in the appendix after the paper is accepted.
> | Model |  Music D1 | Music D2 | Social D1| Average|
> | :--------: | :--------: | :--------: | :--------: | :--------: |
> | RoBERTa | 57.34 | 70.86 | 63.29 | 63.83$_\{\pm 0.18}$ |
> | BERT | 54.26 | 67.13 | 62.58 | 61.32$_\{\pm 0.21}$ |
> | Con_pt | 59.43 | 72.04 | 63.97 | 65.15$_\{\pm 0.20}$ |
> | ERNIE-Baidu | 59.43 | 73.60 | 65.88 |  66.30$_\{\pm 0.14}$ |
> | ERNIE-THU | 58.62 | 72.92 | 65.10 | 65.55$_\{\pm 0.17}$  |
> | KnowBERT | 59.91 | 74.84 | 66.36 |  67.04$_\{\pm 0.24}$ |
> | K-BERT | 58.74 | 73.71 | 67.01 | 66.49$_\{\pm 0.19}$ |
> | KGAP | 59.62 | 74.35 | 66.82 | 66.93$_\{\pm 0.22}$ |
> | DKPLM |57.33| 73.64 | 67.43 | 66.13$_\{\pm 0.15}$ |
> | GreaseLM | 60.18| 74.95 | 68.14 | 67.76$_\{\pm 0.20}$ |
> | KALM | 58.94 | 73.55 | 67.85 | 66.78$_\{\pm 0.15}$ |
> | KANGAROO | 61.51 |  75.89 | 69.27 | 68.89$_\{\pm 0.18}$ |
>
> We sincerely thank for your time and efforts.

---

### Meta-Review · Area_Chair_W7Pi · 2023-09-15

**Recommendation:** 5

**Metareview:**

This paper proposes a new representation learning framework for closed-domain tasks, wherein representations are trained to capture implicit graph structure within the modelled data. The framework is shown to lead to improved performance on named entity modelling on two benchmarks.

Beyond simply demonstrating state of the art results, the paper is also interesting as the results may lead to further discussion of what knowledge structures entity embeddings encode. The paper is well-written and clear, and as such provides a good starting point for such a discussion.

---

### Decision · Program_Chairs · 2023-10-07

**Decision:**

Accept-Main

**Comment:**

This paper proposes a new representation learning framework for closed-domain tasks, wherein representations are trained to capture implicit graph structure within the modelled data. The framework is shown to lead to improved performance on named entity modelling on two benchmarks.

Beyond simply demonstrating state of the art results, the paper is also interesting as the results may lead to further discussion of what knowledge structures entity embeddings encode. The paper is well-written and clear, and as such provides a good starting point for such a discussion.